# Discovery of α-amylase and α-glucosidase dual inhibitors from NPASS database for management of Type 2 Diabetes Mellitus: A chemoinformatic approach

Wilberforce Ndarawit[1], Charles Otieno Ochieng[2], David Angwenyi[3], Jorddy N. Cruz[4], Cleydson B. R. Santos[4,5], Njogu M. Kimani[1,6]*

1 Department of Physical Sciences, University of Embu, Embu, Kenya, 2 Department of Chemistry, Maseno University, Maseno, Kenya, 3 Department of Mathematics, Masinde Muliro University of Science and Technology, Kakamega, Kenya, 4 Department of Biological Sciences and Health, Laboratory of Modeling and Computational Chemistry, Federal University of Amapá, Macapá, Amapá, Brazil, 5 Graduate Program in Medicinal Chemistry and Molecular Modelling, Health Science Institute, Federal University of Pará, Belém, PA, Brazil, 6 Natural Product Chemistry and Computational Drug Discovery Laboratory, Embu, Kenya

* njogu.mark@embuni.ac.ke

**Data Availability Statement:** All relevant data are within the manuscript and its Supporting Information files.

## Abstract

Postprandial hyperglycemia, typical manifestation of Type 2 Diabetes Mellitus ($T_2DM$), is associated with notable global morbidity and mortality. Preventing the advancement of this condition by delaying the rate of glucose absorption through inhibition of α-amylase and α-glucosidase enzymatic activities is of utmost importance. Finding a safe antidiabetic drug is essential since those that are currently on the market have drawbacks like unpleasant side effects. The current study utilized computer-aided drug design (CADD), as a quick and affordable method to find a substitute drug template that can be used to control postprandial hyperglycemia by modulating the activity of α-amylase and α-glucosidase enzymes. The Natural Products Activity and Species database (NPASS) (30,926 compounds) was screened *in silico*, with a focus on evaluating drug-likeness, toxicity profiles and ability to bind on a target protein. Two molecules NPC204580 (Chrotacumine C) and NPC137813 (1-O-(2-Methoxy-4-Acetylphenyl)-6-O-(E-Cinnamoyl)-Beta-D-Glucopyranoside) were identified as potential dual inhibitors for α-amylase and α-glucosidase with free binding energies of -14.46 kcal/mol and -12.58 kcal/mol for α-amylase, and -8.42 kcal/mol and -8.76 kcal/mol for α-glucosidase, respectively. The molecules showed ionic, H-bonding and hydrophobic interactions with critical amino acid residues of both enzymes. Moreover, 100 ns molecular dynamic simulations showed that both molecules are stable on the receptors' active sites based on root mean square deviation (RMSD), root mean square fluctuation (RMSF), and the Generalized Born surface area (GBSA) energy calculated. The two compounds are thus promising therapeutic agents for $T_2DM$ that merit further investigation due to their excellent binding energies, encouraging pharmacokinetics, toxicity profiles, and stability as demonstrated in simulated studies.

**Funding:** NMK received a research grant form the International Foundation for Science (IFS –Grant No: I1-F-6556-1). The funder had no role in the study design, data collection and analysis, decision to publish, or preparation of the manuscript.

**Competing interests:** The authors have declared that no competing interests exist.

## Introduction

Postprandial hyperglycemia, commonly referred to as Type 2 Diabetes Mellitus ($T_2DM$), represents a complex metabolic disorder stemming from either dysregulation in insulin sensitivity, insulin secretion, or both [1]. This condition, if not managed properly, may lead to a chronic condition that is manifested by persistent high blood sugar levels, and over time, contributes to the dysfunction of various vital organs such as the heart, blood vessels, nerves, eyes, and kidneys [2]. Statistical indicators pointed out that, as of the year 2017, a whopping 450 million adults had been affected by diabetes, and the number is anticipated to double in the next two decades [3]. The prevalence of the rising sedentary lifestyle and overreliance on processed food are considered significant contributors to the rising number of adults having $T_2DM$ [4]. Starch consumption provides the body with its main source of energy following a series of digestive enzymatic activities including α-amylase in the saliva, pancreatic α-amylase, and intestinal α-glucosidase localized in the epithelial layer of the ileum [5]. At the initial phase of starch digestion, α-amylase converts complex polysaccharides into simpler oligosaccharides through the cleaving of α-1,4-glucan linkages. This enzymatic action results in the transformation of complex polysaccharides into more manageable simpler oligosaccharides which are further hydrolyzed by the α-glucosidase in the gut [6].

The common medical strategy used in the management of postprandial hyperglycemia involves inhibition of the starch hydrolytic enzymes; α-amylase and α-glucosidase with either a synthetic or natural compound [1,2,7]. Enzymatic inhibition offers a workable first-line $T_2DM$ treatment by decreasing the rate of starch hydrolysis and preventing sudden spikes of glucose resulting in manageable postprandial hyperglycemia [8]. Acarbose, miglitol, and voglibose are examples of contemporary α-amylase and α-glucosidase inhibitors, however these medications suffer certain limitations including unfavorable side effects such as diarrhea, hypoglycemia, liver damage, stomach discomfort, and heart-related complications [7,9,10]. Therefore, a safer dual competitive inhibitor that can inhibit both α-amylase and α-glucosidase active sites would be the optimal alternative approach to provide more effective management of postprandial hyperglycemia.

Every drug discovery effort either through synthetic or extraction from natural sources aims to identify a drug candidate with the best biological performance. However, conventional methods of drug development are associated with varied demerits such as time consumption and the expensive nature of the process [11]. Consequently, *in-silico* approaches plays significant role in the initial stages of drug template identification from myriad preestablished molecular entities as it helps to overcome some of the limitation of conventional drug discovery process [12,13]. In the realm of natural products, the integration of Artificial intelligence (AI) and Computer aided drug design (CADD) techniques has revolutionized the landscape of drug discovery. It has the capacity to scrutinize vast chemical databases and extract meaningful patterns, thereby significantly reducing the time involved in identifying potential drug candidates [14]. With a plethora of databases having plant derived compounds, AI serves as a powerful tool in scouting these databases for novel bioactivities, thereby expediting the drug discovery process, especially in phytomedicine in which application of AI tools holds immense promise for identifying potential therapeutic agents [15].

Therefore, in this study AI techniques such as virtual compound screening, molecular modeling, molecular docking and molecular dynamics simulation, pharmacokinetic properties and toxicity evaluation have been employed in finding a probable dual inhibitor for α-amylase and α-glucosidase receptors. Thereby by harnessing AI driven methodologies, Virtual screening was conducted on 30,926 compounds from the Natural Products Activity and Species Source Database (NPASS Database (bidd.group)) using the molecular operating

environment (MOE) software, Data Warrior and pkCSM (pkCSM (uq.edu.au)) webpage for toxicity screening to establish suitable non-toxic drug candidates.

## Materials and methods

### Virtual screening

Structure-Based Drug Design (SBD) was adopted in this method of screening the NPASS database to find an alternative inhibitor that can be used to control $T_2DM$. Step-by-step cutting-edge cheminformatics techniques, including *in-silico* drug-likeness screening, pharmacokinetics, toxicity, molecular docking, and molecular dynamic simulation were employed to screen the curated NPASS database in finding possible dual inhibitors for α-amylase and α-glucosidase enzymes.

### Retrieval and preparation of ligands for docking

A total of 30,926 molecules used in the study, were sourced from the NPASS (NPASS Database (bidd.group)). Since its initial release in 2017, the NPASS has been one of the most important data source for natural product research community [16]. These molecules were accessed on December 20th, 2022, and downloaded in SDF format. The 3D molecular structure of acarbose was acquired from PubChem in SDF format. Acarbose was used as a reference molecule owing to it being co-crystalized with both α-glucosidase (2QMJ) and α-amylase (3BAJ) proteins. All the molecules in the form of an SDF file were then exported into Molecular Operating Environment (MOE) 2015.10 software in which 3D representations of respective ligands were generated. The molecules underwent curation by washing to remove duplicate molecules and molecules with errors or inconsistencies in their structure. Additionally, protonation at 300 K and a pH of 7 was done so that the necessary hydrogen atoms were added to ensure the molecular structures were accurately represented. Subsequently, the energies of the molecules were minimized using MOE software, employing the default parameters for the computation and energy minimization of the molecules (MOE: compute, molecules, energy minimize).

### Drug-likeness and in-silico pharmacokinetics screening

Beyond Lipinski's Rule of Five (bRO5) was used to characterize and screen the Pharmacokinetics and drug-likeness properties of the compounds following MedChem Mnemonic Rules such as Lipinski, Verber, Shultz, and Meanwell [17–20]. The database was screened using the tenets of bRO5 including, Molecular weight (MW) range from 260 to 500, Topological polar surface area (TPSA) ranges from 75–140 $Å^2$, Hydrogen bond acceptors (HBA) $\leq$ 10, Hydrogen bond donors (HBD) $\leq$ 5; number of rotatable bonds (RT) $\leq$ 10; and hydrophobicity (log P) $\leq$ 5, Heavy atoms count (HA) < 25, and total HBA + HBD $\leq$12. The MOE software was modified to screen non-drug-like molecules using the aforementioned criteria, as generated by the MOE software (MOE, Compute, descriptors;(MW, HBA, HBD, TPSA, HA, RT and log P), calculate).

### Toxicity based screening

The toxicity parameters of the compounds that were retained after performing *in-silico* pharmacokinetics and drug-likeness screening were determined using pkCSM webserver (pkCSM (uq.edu.au) [21] and Data Warrior software [22]. The toxicity metrics acquired from the pkCSM comprised AMES toxicity, hERG I & II inhibitor, maximum tolerant dose (human) (MTD), Oral Rat Acute Toxicity $LD_{50}$, Minnow toxicity, and skin sensitization (SS). Data Warrior was also used to determine compounds that were mutagenic, tumorigenic,

reproductive-effective, and irritating molecules. All the toxicity parameters mentioned were assessed for the compounds identified after doing drug-likeness and *in-silico*-pharmacokinetics screening.

## Retrieval and preparation of receptor/protein

The 3D Crystal Structure of Catalytic α-glucosidase (2QMJ) and Human Pancreatic α-Amylase (3BAJ) in Complex with acarbose were retrieved from Protein Data Bank (PDB) on May 2, 2023 [23,24]. Examination of the ligand interaction of the co-crystalized ligand was carried out to determine the protein's active site, eliminate any water molecules, and residual ligand (MOE: SEQ), and correct any missing amino acid residues. (Compute, Prepare, Structure preparation, Correct). The protein model's molecular systems were then protonated, ionization states assigned and hydrogen atoms positioned using the MOE software at 300 K and pH 7.0. Finally, the models' energy system was minimized to obtain a relaxed receptor for molecular docking (MOE, Compute, Energy Minimize, and Tether Atoms), followed by tethering of atoms to ensure that the receptor structure established by experimentation did not significantly vary with the prepared receptor.

## Molecular docking

The remaining compounds upon drug-likeness, and toxicity screening, underwent screening utilizing molecular docking methodologies, in which they were docked into the active sites of proteins α-amylase (3BAJ) and α-glucosidase (2QMJ). Subsequently, a screening was performed based on their respective docking scores. Prior to docking, the receptor active sites for each protein model consisting of Trp59, Tyr62, Gln63, Thr163, Arg195, Asp197, Lys200, His201, Glu233, Glu240, and Asp300 for α-amylase and Thr205, Asp203, Met444, Asp542, Asp443, Asp327, Arg526, and His600 for α-glucosidase were carefully chosen and occupied by dummy atoms of molecules. The MOE-Dock module with the following parameters: placement-Triangle matcher, rescoring-London dg, refinement-rigid receptor was set for docking. The conformers of respective ligands were subjected to docking within a designated binding pocket of respective enzymes; 3BAJ and 2QMJ and each molecule's ideal positioning/orientation within the respective receptor pocket were calculated using MOE software. The resulting root mean square deviation (RMSD) values and binding affinities of the highest-ranked molecule having the highest binding affinity (the most negative) were used to investigate the binding interactions of the ligands with 2QMJ and 3BAJ proteins. Through conformational sampling, the best binding modes for the identified molecules were determined. The Protein-Ligand Interaction Fingerprints (PLIF) feature in MOE software was utilized to visualize binding modes between the receptor's binding pocket and the ligand molecule with the best docking score. The software calculates various types of interactions, such as hydrogen bonds, water-mediated protein-ligand interactions, ionic interactions, surface contacts, metal binding, and aromatic interactions between the ligand and the receptor, all in 2D format.

The molecular docking approach that was utilized in this research was validated using the acarbose (a co-crystalized ligand and known inhibitor for the two enzymes) before docking was done on the two enzymes with the screened NPASS molecules. The acarbose was subsequently re-docked to the original site where the co-crystal ligand initially interacted with the corresponding receptor. The experimental conformational (the crystallographic pose) and the postures produced by docking with the reference acarbose were superposed and compared (Fig 1). The conformer with highest binding affinity and having a comparable posture with a co-crystalized ligand from the validation docking for each receptor was selected for each enzyme structure and served as the standard for all succeeding docking simulations with the two proteins.

**3BAJ** **2QMJ**

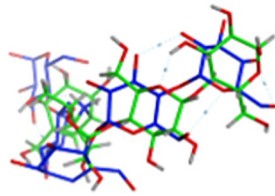 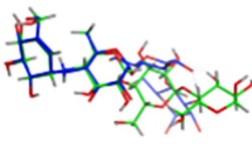

**Fig 1. Superposed structures α-amylase (3BAJ) and α-glucosidase (2QMJ).** The most favorable docking pose of 3BAJ and 2QMJ is represented in blue and the crystallographic pose shown in green within superimposed structures of acarbose after the binding interaction was validated with the help of MOE software.

## Molecular Dynamic (MD) simulations

MD simulation of the four complexes was performed using Amber 16 [25]. The partial atomic charges of the ligand were determined using the electrostatic potential (RESP) [26] protocol carried out in the Gaussian 09 program using Hartree-Fock level with the 6-31G* basis set [27]. The biomolecular systems were solvated in an octahedral periodic box, with TIP3P water model [28]. The cutoff between the water box wall and the atoms of the solvated system was set to 12 Å. Counter ions $Na^+$ was added to the system to maintain electroneutrality. The force fields 14SB [29] and the General Amber Force filed (GAFF) [30] were employed in parameterizing the biomolecular systems as well as the ligand structures respectively. Energy minimization protocol was performed for all the systems through six steps with steep-descent and conjugate gradient algorithm. This was performed following a series of 10,000 cycles for each step to minimize the hydrogen atoms, water and ions and completely minimizing the whole system with the progressive decrease of restrains. Next, the system was heated from 0 to 300 K over 800ps in three stages and maintained at 300 K by coupling to a Langevin thermostat using collision frequency of 2 $ps^{-1}$ and a constant pressure of 1 bar. Equilibration was performed over 2 ns at constant temperature without any restrains. Subsequently, 100 ns of MD simulation was carried out for each complex using the NVT ensemble. Ligand Root mean square deviation (RMSD) and receptor Root-mean-square fluctuation (RMSF) in each complex system were calculated using the MD simulation trajectories.

## Energy minimization calculations

The binding free energy of the ligands in the receptor was determined using mechanics-generalized surface area (MM-GBSA) method [31]. 500 snapshots of the last 5ns of the MD trajectories were used in determining the free binding energy according to the following equations:

$$\Delta Gbind = \Delta H - T\Delta S \approx \Delta EMM + \Delta Gsolv - T\Delta S$$

## Results and discussion

### Drug-likeness and in-silico pharmacokinetics screening

Virtual screening (VS) (summarized in Fig 2) is a crucial, often applied procedure in drug development while exploring prospective lead molecules that can inhibit or trigger biomolecular receptors [32].

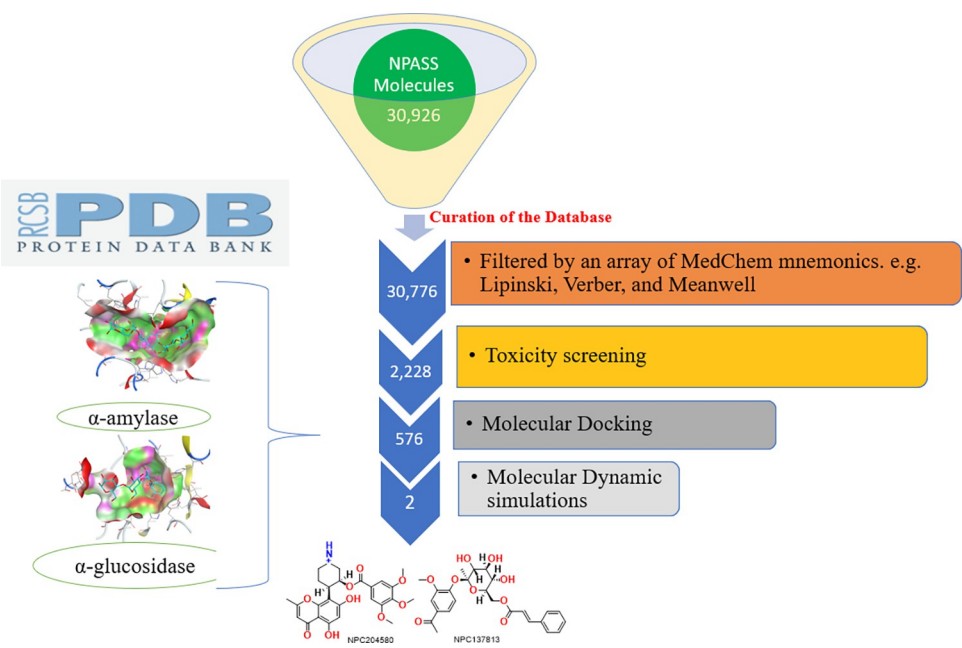

**Fig 2. Schematic flow chart of structure-based virtual screening of the 30,926 Natural Products Activity and Species Source Database (NPASS) compounds.**

A compound's potential as a therapeutic agent is largely dependent on its molecular physicochemical properties as predisposing conditions for drug-likeness [33,34]. Beyond Lipinski rule of five (bRO5) was utilized in screening 30,926 molecules from the NPASS database based on physicochemical properties to have drug-like molecules with favorable pharmacokinetic properties, emphasizing compounds that meet the requirements with zero violations of bRO5.

The curation process of NPASS database eliminated 150 molecules from the initial set of 30,926 molecules, specifically removing duplicate entries and incomplete or incorrect data ensuring a refined selection of relevant, quality candidates for further virtual screening.

A further curation following the rules delineated by MedChem mnemonics such as Lipinski, verber and Meanwell resulted in 2,228 molecules; their physicochemical properties are summarized in (S1 Fig). According to Lipinski, a molecule having a molecular weight (MW) falling within the range of 260 to 500 Daltons often exhibit desirable absorption, distribution, metabolism, and excretion (ADME) properties which are essential in ensuring effective drug metabolism [35]. The partition coefficient between n-octanol and water (Log $P_{o/w}$) value is crucial for evaluation of a compound's lipophilicity, which determines its solubility and ability to pass through biological cell membranes [36]. Appropriate Log P for good drug-like molecules should be <5 to enhance the equilibrium between hydrophilicity and lipophilicity, which is vital for bioavailability of the molecule [37]. The number of heavy atoms (HA) and the number of rotatable bonds play a crucial role in the compound's stability, flexibility and interactions with the biological target [38]. Molecules that surpassed the metrics of HA< 26 and ≥10 rotatable bonds fail to meet the bRO5. Topological polar surface area (TPSA) is a crucial factor in the selection of drug candidate during virtual screening as it has effects on drug permeability, bioavailability, solubility and receptor interactions [39,40]. Therefore, druglike molecules with TPSA values ranging between 75 Å$^2$ to 140 Å$^2$ were considered to be prospective drug candidates as they could have an optimal surface area (SA) to interact with the target receptor, but have no effect on membrane permeability [39,40]. Hydrogen bond acceptors

(HBA) and donors (HBD) plays a crucial role in determining compound's ability to interact with specific target receptor [17]. As such, HBA is set at ≤10 while HBD at ≤5 to ensure the optimal biochemical interactions including hydrogen bonding interactions that are essential for biological activity, as their presence influences the capacity of a molecule to establish chemical interactions with specific proteins or receptors.

Therefore, the physicochemical parameters as summarized in (S1 Fig) demonstrates that all the compounds are potential druglike molecules. However, further pharmacokinetics and toxicological analysis offered additional screening criteria.

## Toxicity based screening

Toxicity-based screening is an essential procedure in the pharmaceutical industries in assessing the safety profiles of molecules [41]. The process entails a thorough assessment of substances to determine any potential hazards or adverse effects they may have on human health. In this study pkCSM webserver and Data-warrior were used to evaluate and screen the molecules based on various toxicity endpoints such as AMES toxicity, hERG Inhibitor, Hepatotoxicity, and Skin Sensitization, tumorigenic, reproductive-effective, and irritating molecules, the toxicity profile of the molecules are given in Fig 3(A) and 3(B).

The AMES toxicity parameter is a reliable method for predicting the mutagenicity and carcinogenic potential of chemical substances [42]. It primarily identifies compounds that can cause genetic mutations, leading to cancer or in-hereditable genetic disorders [43]. Out of the 2228 molecules analyzed using pkCSM, 36.5% exhibited AMES toxicity which concurred with results from Data Warrior showing 38.2% of the molecules showed as mutagenic, indicating a high likelihood of causing genetic mutations in an organism. Consequently, these identified compounds were eliminated because of their capacity to induce detrimental impacts on human health.

The primary factor responsible for the emergence of acquired long QT syndrome, which can result in fatal ventricular arrhythmia or other severe cardiac ailments, is the suppression of potassium channels encoded by hERG (human ether-a-go-go-gene) [44]. A number of drugs have been withdrawn from the pharmaceutical market due to their potential to inhibit hERG channels. Therefore, in the virtual screening 20.1% of the molecules that exhibited hERG inhibition were eliminated.

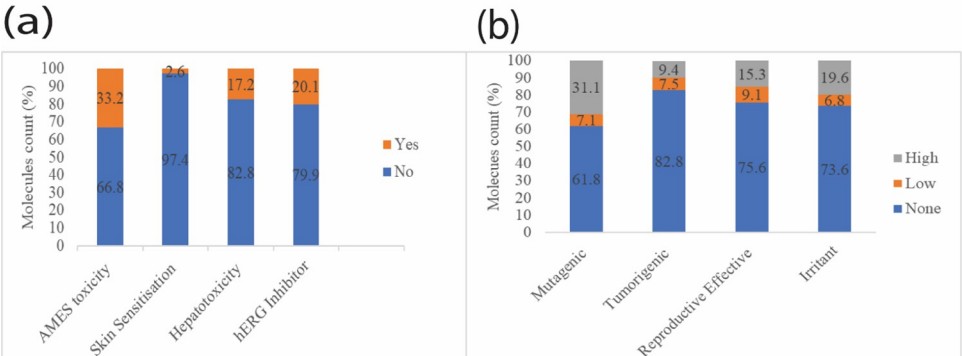

**Fig 3. Graphs of toxicological properties of the 2228 screened molecules.** (a) stack bar graphs showing the percentage AMES toxicity, Skin Sensitization, Hepatotoxicity and hERG inhibitors determined using the pkCSM, (b) Stacked bar graphs showing Mutagenic, Tumorigenic, reproductive effective and irritant molecules determined using Data Warrior.

Hepatotoxicity screening is essential as it assesses the molecule's potential harm to the liver, a vital organ responsible for detoxification and metabolism of chemical substances and nitrogenous waste in the body [45]. The toxicological analysis eliminated up to 17.2% of molecules that exhibited hepatotoxic properties during screening.

Similarly, the assessment of the skin sensitization (irritant screening) predict the capacity of a drug to elicit allergic reactions when it comes into contact with the skin [21]. Out of the compounds analyzed with pkCSM, only 2.6% of the compounds exhibited skin sensitization, whereas 26.4% (19.6%-High and 6.8%-Low) of the compounds were found to have the potential to cause general allergic reactions based on Data Warrior results. Therefore, these compounds were excluded to mitigate the potential for allergic skin reactions, with a focus on promoting the development of a safer drug.

Tumorigenic screening is a process that seeks to detect substances that have the potential to cause the development or expansion of tumors [46]. This assessment is important in order to prevent exposure to possible cancer-causing agents. Reproductive-effect evaluations prioritize assessing the possible negative effects on fertility, pregnancy, or the development of the fetus, to guarantee the safety of drugs. Molecules that exhibited any form of toxicity based on tumorigenic and reproductive effectiveness were eliminated during the screening process.

Following this extensive screening process, the pool of prospective therapeutic candidates was reduced to 576 compounds that satisfied the safety criteria. These molecules were further subjected to molecular dockings to predict their binding interactions with target proteins, aiding in understanding their mechanism of action and potential as therapeutic agents.

## Docking studies

Molecular docking serves as a powerful computational technique for finding probable chemical entities that can be utilized in investigating the binding potency of chemical molecules to a particular target such as enzymes and nucleic acids [47–49]. The molecular docking technique was employed in exploring detailed molecular docking studies of the 576 NPASS compounds and the respective target receptors (2QMJ and 3BAJ) using MOE software. Binding interaction residues, root mean square deviation, and docking score were used to grade compounds in comparison to the reference ligand Acarbose, and the results of the top ranked molecules in the two receptors are summarized in (S1 and S2 Tables).

## Binding interaction analysis of α-amylase (3BAJ)

The active site of the Human pancreatic α-amylase (Fig 4), containing the selected amino acid residues Trp59, Tyr62, Gln63, Thr163, Arg195, Asp197, Lys200, His201, Glu233, Glu240, and Asp300, was docked with the optimized 576 screened molecules from the NPASS database.

Upon docking the docked molecules were screened based on the docking threshold of -10.0 kcal/mol and less ≤2.0 Å RMSD values, as these were considered the most stable ligand in the pocket of the receptor. Four ligands (NPC204580, NPC137813, NPC76084, and NPC27750) were regarded as the most probable inhibitors of 3BAJ protein among the screened derivatives due to their high binding affinities, and outstanding ligand interactions with the designated amino acid, in addition to low RMSD values. The selected molecules' binding interactions, docking scores, and RMSD values were analyzed and the findings are shown in (S1 Table). It is noted that all the four ligands formed more than three hydrogen bond interactions with the amino acid, as indicated in S1 Table. However, two of the ligands (NPC204580 and NPC137813) exhibited docking scores comparable to Acarbose, whereas NPC76084 and NPC27750 displayed significantly low binding affinities relative to the reference compound Acarbose. Therefore, NPC204580 and NPC137813 were considered for further studies as

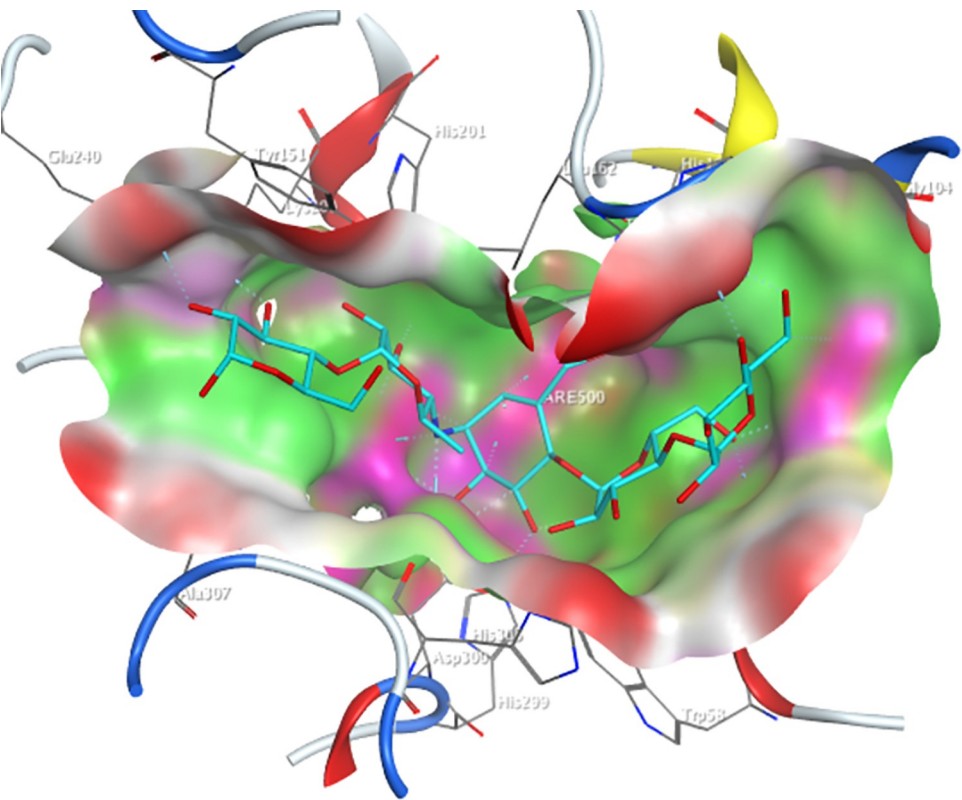

**Fig 4. Active-site of the human pancreatic α-amylase with co-crystalized molecule acarbose (cyan).**

potential inhibitors based on this rationale. The NPC204580 molecule formed an ionic bond with Asp 197 and a 6-ring pi bond interaction with His 201 residues, alongside three hydrogen bonds with Asp300, Glu233 and Lys200, hence accounting for its high binding affinity of -14.46 kcal/mol docking score. NPC137813 on the other hand formed four hydrogen bonds with Asp 300, Asp 197, His 299, and Arg 195 amino acid residues which also gave it a high binding affinity of -12.58 kcal/mol.

## Binding interaction analysis of α-glucosidase (2QMJ)

The selected structures of NPASS molecules were also docked into the determined active site of the human maltase-glucoamylase receptor (PDB ID: 2QMJ) (Fig 5).

The derivatives NPC25750, NPC137813, NPC76084, and NPC204580 of the NPASS database demonstrated docking scores ranging from -8.42 kcal/mol to -9.08 kcal/mol, closely aligning with the docking score of the reference compound Acarbose (-8.22 kcal/mol). Furthermore, the ligands exhibited exceptional binding affinity towards key amino acid residues of human pancreatic α-glucosidase, including; Thr205, Asp203, Met444, Asp542, Asp443, Asp327, Arg526, and His600 (S2 Table). These interactions contribute to their high binding affinities.

The docking scores and root mean square deviation (RMSD) of the four ligands and the reference compound Acarbose were analyzed and compared in both receptors as illustrated in Fig 6 (6a_3BAJ and 6b_2QMJ). The comparison of these parameters highlights the relative binding efficacy and structural alignment of the ligands with the standard drug acarbose, offering critical insights into their potential inhibitory activity. The results revealed that the selected

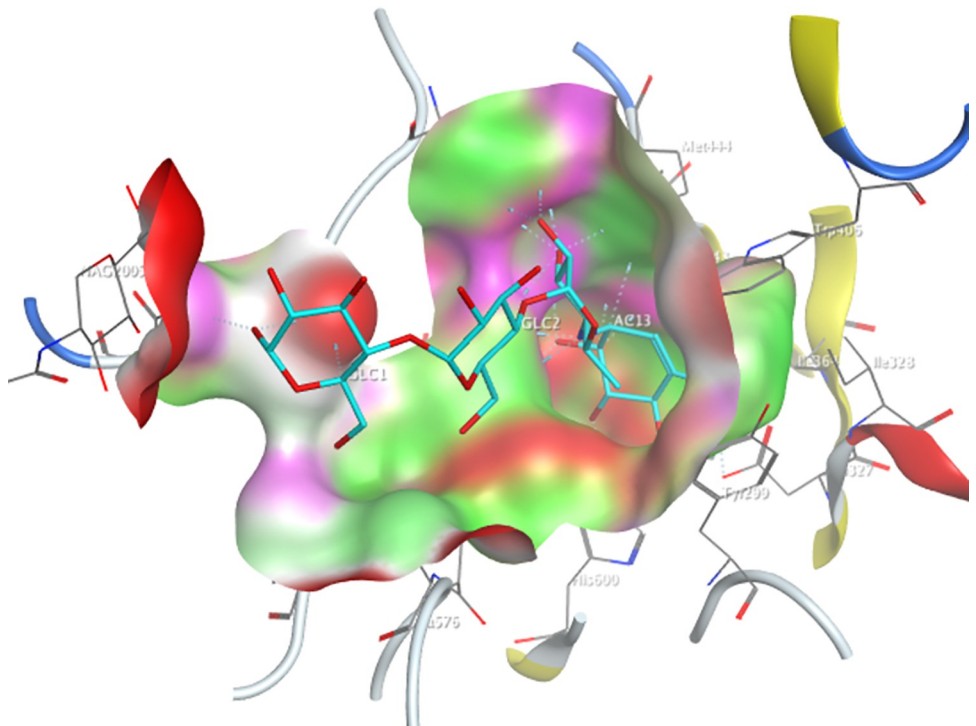

**Fig 5. Active-site of the human pancreatic α-glucosidase with co-crystalized molecule acarbose (cyan).**

four ligands had demonstrated significantly comparable docking scores with acarbose in α-glucosidase. However, NPC76084 and NPC25750 displayed scores below the threshold in α-amylase interaction in comparison with acarbose. Therefore, such reasoning left only NPC204580 and NPC137813 as dual inhibitors of the two enzymes.

## Molecular docking studies of the selected dual inhibitors and acarbose with α-amylase and α-glucosidase

Molecular docking studies of the selected dual inhibitors and the Acarbose (reference) revealed that all ten poses demonstrated comparable docking scores with Acarbose as illustrated in S3 Table and Fig 7. The two molecules NPC204580 and NPC137813 with α-glucosidase revealed that all of their ten poses demonstrated lower docking score (S-score) than the Acarbose (Fig 7B), suggesting that the two molecules possess stronger binding affinity towards the

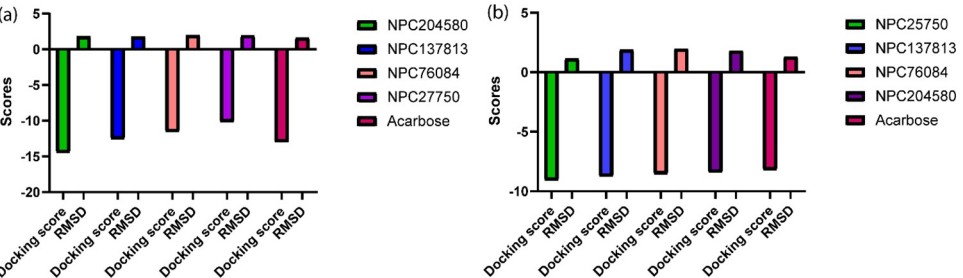

**Fig 6.** Docking scores and RMSD bar graphs for selected ligands in both α-amylase (a) and α-glucosidase (b) receptors.

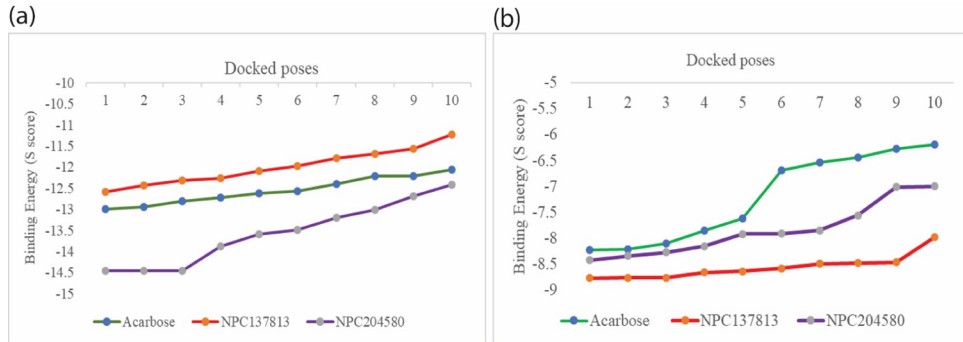

**Fig 7.** Top ten poses of NPC204580, NPC137813, and Acarbose on the x-axis plotted against the binding affinity (S-score) for alpha-amylase (a) and alpha-glucosidase (b). The greater the negative value of S-score, the stronger the predicted binding affinity.

α-glucosidase, potentially leading to more effective inhibition of the α-glucosidase. Additionally, NPC204580 exhibited a lower S-score than acarbose with α-amylase, suggesting a better binding affinity, but the S-score of NPC137813 was higher than that of reference molecules acarbose in all the ten poses with α-amylase (Fig 7A), however, the difference was comparable. Therefore, both NPC204580 and NPC137813 were considered a promising therapeutic candidate against T$_2$DM due to their favorable binding affinities with α-amylase and α-glucosidase which are key targets for inhibiting carbohydrates digestion and glucose absorption in the gut.

### Binding interaction analysis of the selected two dual inhibitors

A detailed examination of the binding interactions of the best-docked poses in the two ligands NPC204580, NPC137813, and Acarbose with the receptors: 3BAJ and 2QMJ was conducted. This analysis revealed significant binding interactions as illustrated in Table 1, along with their docking scores, and RMSD values. The estimated binding free energy for NPC204580 and NPC137813 with 3BAJ was -14.46 kcal/mol and -12.58 kcal/mol, respectively while in 2QMJ it

**Table 1. Docking score, RMSD values, and the interactions of selected dual inhibitors ligands and acarbose with the receptors 3BAJ and 2QMJ residues revealed during structure visualization.**

| α-amylase (ID: 3BAJ) | | | | | α-glucosidase (ID: 2QMJ) | | | |
|---|---|---|---|---|---|---|---|---|
| Compound | Docking score Kcal/mol | RMSD values (Å) | Binding resides | Interaction | Docking score Kcal/mol | RMSD values (Å) | Binding resides | Interaction |
| NPC204580 | -14.46 | 1.80 | Asp 300<br>Glu 233<br>Lys 200<br>Asp 197<br>His 201 | H-donor<br>H-donor<br>H-acceptor<br>Ionic<br>pi-H | -8.42 | 1.79 | Met 444<br>Asp 542<br>Asp 203<br>Asp 542<br>Phe 575 | H-donor<br>H-donor<br>Ionic<br>Ionic<br>H-pi |
| NPC137813 | -12.58 | 1.79 | Asp 300<br>Asp 197<br>His 299<br>Arg 195 | H-donor<br>H-donor<br>H-acceptor<br>H-acceptor | -8.76 | 1.88 | Thr 205<br>Arg 202<br>Thr 204 | H-acceptor<br>H-acceptor<br>Pi-H |
| Acarbose | -12.99 | 1.61 | Gln 63<br>Trp 59 | H-donor<br>H-pi | -8.22 | 1.29 | Asp 203<br>Asp 542<br>Asp 327<br>Met444<br>Asp 443<br>Arg 526<br>His 600<br>Asp 542 | H-donor<br>H-donor<br>H-donor<br>H-donor<br>H-donor<br>H-acceptor<br>H-acceptor<br>Ionic |

demonstrated -8.42 kcal/mol and -8.76 kcal/mol, respectively. Additionally, the RMSD values of the two ligands were less than 2 Å in each receptor. Additionally, the RMSD values of the two ligands were less than 2 Å in each receptor, suggesting structural stability of the binding predictions.

Further, visualization of the 2D (Fig 8) and 3D (Fig 9) interaction of the two molecules with the two receptors revealed that, NPC204580 molecule formed five interactions at Asp300, Glu233, Lys200, Asp197, and His200 amino acid residues of the α-amylase Fig8(a), and five interactions at Met444, Asp542, Asp203, Asp542, and Phe575 amino acid residues of α-glucosidase Fig 8(D). The interaction was supported by RMSD value of 1.8 Å and docking score of -14.46 kcal/mol in α-amylase together with two hydrogen bonds with Asp300 and Asp197 residues via the C atom and NH$^+$ group. Moreover, an ionic interaction at Asp197, pi-H interaction to His201 alongside H-acceptor interaction to Lys200, summarized the ligand's overall binding affinity with α-amylase. Interaction of the same ligand (NPC204580) with α-glucosidase was also examined and it revealed that -8.42 kcal/mol docking score, 1.79 Å RMSD was as a result of important interaction between the ligand and the receptor. With 2D interaction studies, two prominent hydrogen bonds at Met444 and Asp542, two ionic interactions with Asp203 and Asp542, and H-pi interactions at Phe575 residues was observed. These interactions of the ligand in both receptors justified the high binding affinity of the ligand and low RMSD values observed with the two enzymes.

The ligand NPC204580 named Chrotacumine C is a naturally occurring chromone alkaloid isolated from leaves and barks of *Dysoxylum acutangulum* where it was established to be active against cancer cell lines [50]. Moreover, the compound was discovered to inhibit Human Neutrophil Elastase (HNE), owing to its significant role in acute respiratory distress syndrome (ARDS), and various inflammatory disorders such as atherosclerosis and dermatitis [33]. Therefore, the compound Chrotacumine C is noted to be a potential drug candidate that can be structurally optimized further for the management of various pathophysiological conditions such as postprandial hyperglycemia.

The ligand NPC137813 with docking score of -12.58 kcal/mol and 1.79 Å RMSD value with α-amylase formed four hydrogen bond interactions at Asp 300, Asp 197, His 299, and Arg 195 amino residues of α-amylase via OH groups Fig 8(B). In addition to, it also portrayed an excellent docking score with less than two RMSD values with α-glucosidase (-8.76 kcal/mol and 1.88 Å, respectively). The ligand formed two hydrogen interactions at Thr 205, and Arg 202 via OH and carbonyl group respectively, alongside Pi-H interaction at Thr204 amino acid residues via a 6-ring molecule Fig 8(E). Therefore, the interactions of these ligands with the respective receptor and low RMSD value in both receptors justifies its binding affinity with the respective enzymes; α-amylase (-12.58 kcal/mol) and α-glucosidase (-8.76 kcal/mol).

The molecule NPC137813 was previously isolated from *Picrorhiza scrophulariiflora* alongside other ten compounds and was established to be good antimalarial compound [51]. However, there is paucity of information detailing the inhibitory activity of this compound against α-amylase and α-glucosidase receptors and the study has thus proved its potential which then requires further optimization for management of postprandial hyperglycemia. In addition to the discussed amino acid residues, the two ligands NPC204580 and NPC137813 formed several additional bond interactions with important catalytic amino acids of the receptor as evident in Fig 8, which will play a considerable role in interfering with the physiological normal functioning of these two enzymes.

Acarbose molecule was used as a control in this study, it displayed a binding score of -12.99 kcal/mol and 1.61 Å RMSD value with α-amylase enzyme and formed hydrogen bond with Gln 63 residue and H-pi interaction with Trp 59 amino acid residue Fig 8(C). An RMSD value of 1.29 Å and a binding score of -8.22 kcal/mol were observed with α-glucosidase. The binding

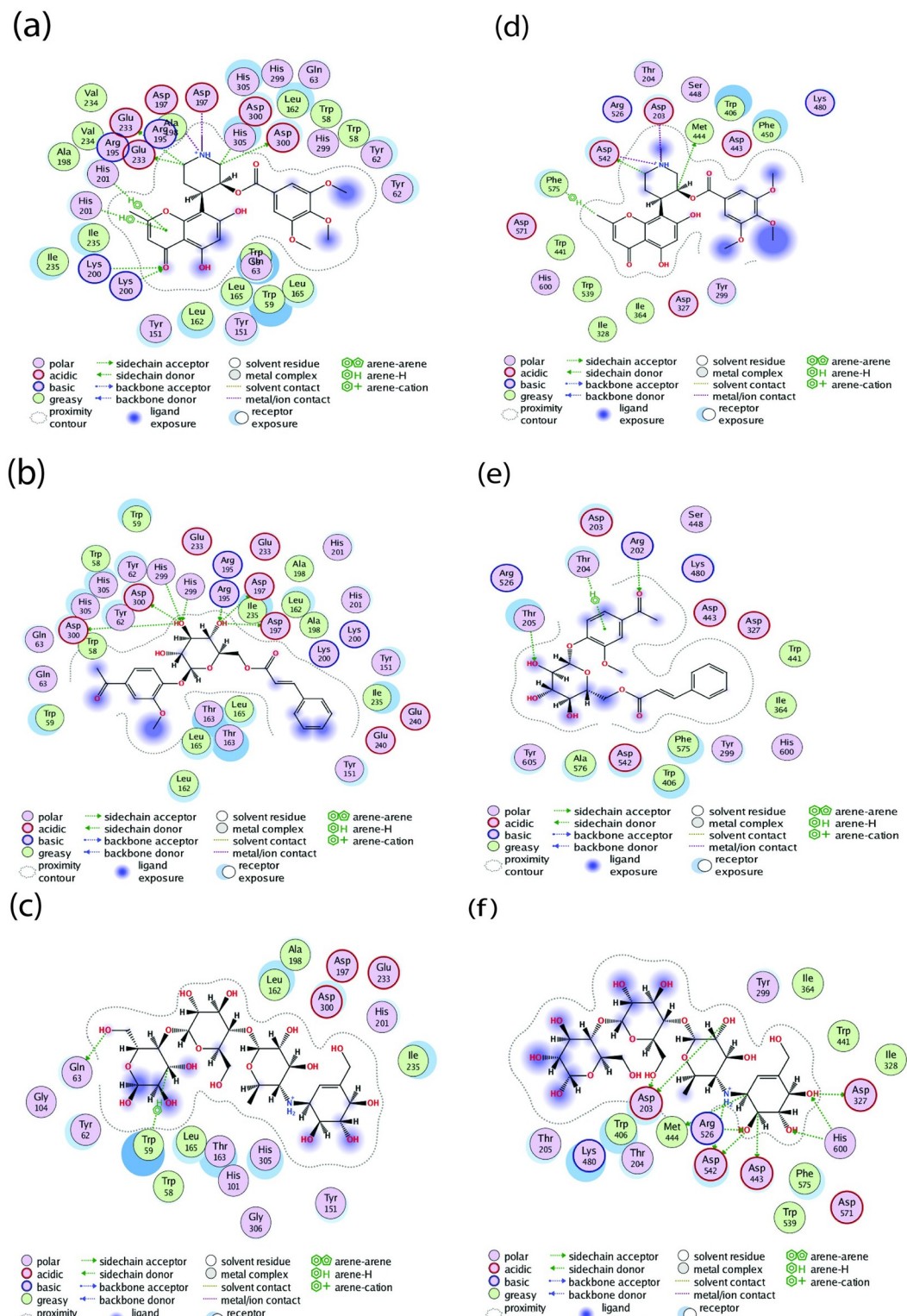

**Fig 8.** 2D interaction of the selected two ligands and the reference molecule; (a)NPC204580_3BAJ, (b) NPC137813_3BAJ, (c) Acarbose_3BAJ, (c) NPC204580_2QMJ, (e); NPC137813_2QMJ and (f) Acarbose_2QMJ.

pocket amino acid residues form hydrogen bonds with Asp 203, Asp 542, Asp 327, Met 444, Asp 443, Arg 526, and His 600 in addition to their ionic interaction with the amino acid residue Asp 542 Fig 8(F). The presence of a greater number of hydroxyl functional groups in the co-crystalized Acarbose structure appears to be the cause of the increased number of hydrogen bonds observed between the ligand and the active site of the receptor [52].

The findings from the docking studies indicate that the chosen ligands exhibit robust binding to the active sites of both receptors. These ligands established strong interactions with crucial catalytic amino acid residues, and their binding affinities are favorable, as seen by RMSD values below 2Å. Therefore, the two inhibitors identified (Fig 10) in this study, are considered to be able to disrupt a crucial carbohydrate metabolic process and may serve as a starting point for structural optimization in finding more effective inhibitors of carbohydrate metabolism hence managing hyperglycemic conditions.

## Pharmacokinetics properties of NPC204580 and NPC137813

A powerful molecule needs to be concentrated sufficiently to reach its target in the body and remain there in bioactive form long enough for the anticipated biological activity to take place for it to be effective as a medication [53,54].

Bioavailability assessment of the final two proposed compounds was computed and evaluated using the Swiss ADME website (SwissADME) [53] and the results are presented in Table 2, Figs 11 & 12. Gastrointestinal absorption and cerebral penetration of the selected molecules were predicted based on the BOILED-Egg plot (Fig 11), which illustrates the relationship between WLOGP and TPSA [55]. The two compounds are projected to have significant GI absorption, none of them have BBB permeation characteristics implying they are highly absorbable in the gastrointestinal tract (IG) (white region), but cannot cross the blood-brain barrier (BBB). Given that the selected molecules are substrates of P-glycoprotein (P-gp) (Table 2), we anticipate that they will be effluxed back into the gastrointestinal tract, hence localizing their activity within the intestinal lumen, while minimizing systemic absorption. Additionally, it has been reported that drug molecules demonstrating a greater negative Log $K_p$ easily permeate through the skin [56]. Therefore, the proposed compounds would justify having transdermal efficacy, as evidenced by the skin permeability of -6.85 cm/s and -8.56 cm/s for NPC204580 and NPC137813 respectively.

The Bioavailability radar (Fig 12) predicted an ideal range of properties for the proposed molecules, the pink zone signifies optimal zone for desired characteristics. This representation suggests that molecules falling within this pink region possesses attributes that align with the features of a molecules to qualify as therapeutic candidate. Therefore, the proposed molecules exhibited favorable bioavailability, indicating their potential effectiveness and suitability for further exploration.

The metabolism and biotransformation of the drug are regulated by a number of cytochromes P450 isoenzymes, including CYP1A2, CYP2C19, CYP2C9, CYP2D6 and CYP3A4 [57]. Therefore, comprehending the significance of the cytochrome P450 system in drug metabolism is essential for understanding drug interactions that may result in toxicities and reduced therapeutic efficacy. The two molecules were noted non-inhibitors of any of the isoenzymes: CYP1A2, CYP2C19, CYP2C9, CYP2D6, and CYP3A4 (Table 2) thus suitable drug candidates.

## Synthetic accessibility

Synthetic accessibility (SA) is an important factor to take into account while selecting most promising lead molecules [58]. SA is evaluated by taking into account factors related to size and complexity such as macrocycles, chiral centers, or spiro functions of a molecule [42]. This

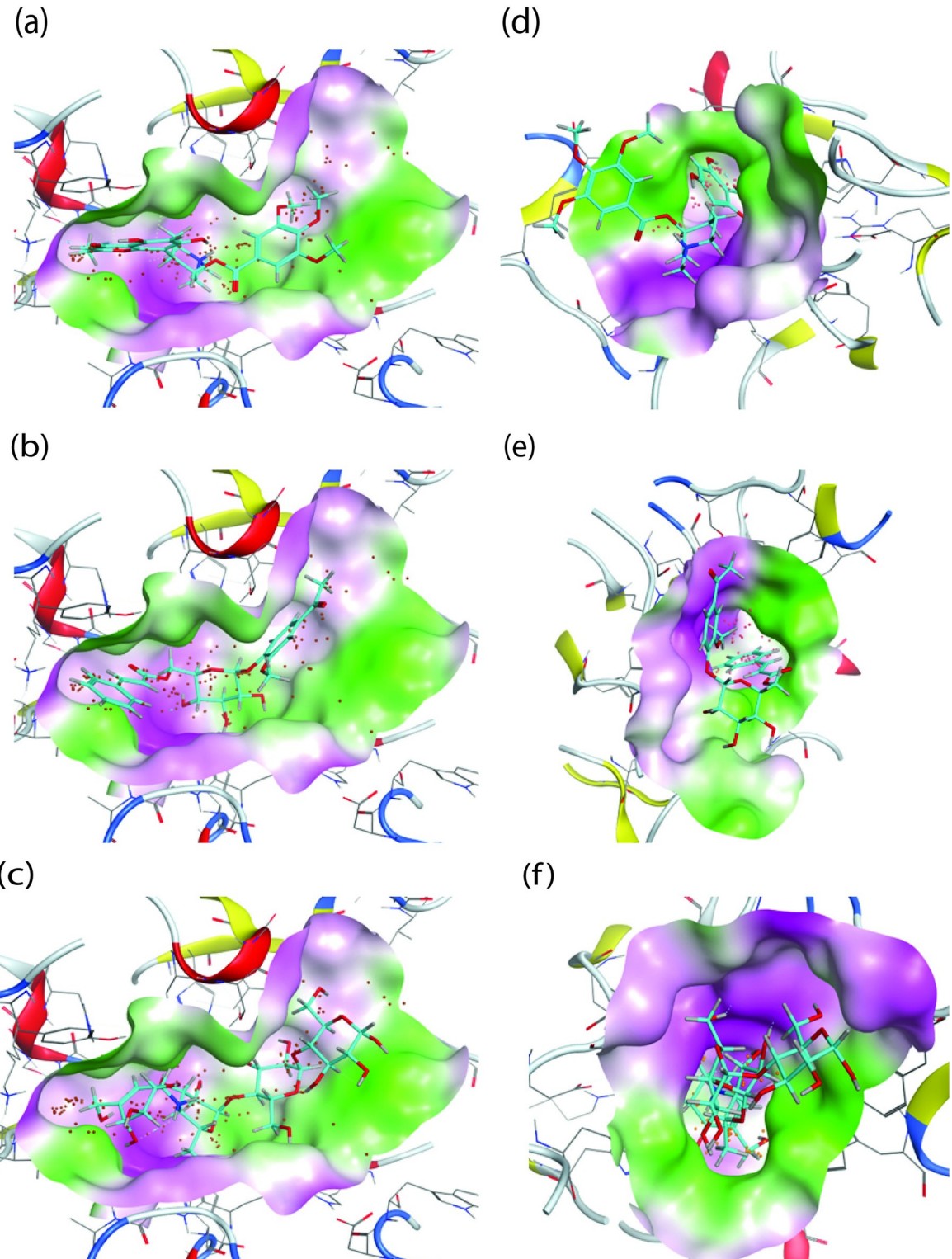

**Fig 9.** 3D Docking conformation of the selected ligands and the reference molecule; (a)NPC204580_3BAJ, (b) NPC137813_3BAJ, (c) Acarbose_3BAJ, (c) NPC204580_2QMJ, (e); NPC137813_2QMJ and (f) Acarbose_2QMJ. The molecular surface is colored green for the lipophilic regions and purple for the hydrophilic region of the receptor.

**NPC204580**                                        **NPC137813**

**Fig 10. 2D structures of the proposed molecules NPC204580 and NPC137813.**

leads to SA scores ranging from 1 (ease to synthesize) to 10 (difficult to synthesize) [53]. Based on the finding shown in Table 2, the two proposed molecules are considered synthetically feasible as their synthetic accessibility score range between 5.30 for NPC204580 and 6.36 for NPC137813. Therefore, the discussed ADME properties further corroborated the good pharmacological attributes and the predicted docking results of the two molecules.

## Molecular dynamic simulations

Molecular dynamic simulation is an essential computational methodology employed in exploring the stability of macro and micro-molecules in a particular water-based environment. In this study, NPC137813 and NPC204580 proposed potential dual inhibitors of α-amylase (3BAJ) and α-glucosidase (2QMJ) receptors were subjected to MD simulation for 100ns. MD simulation, trajectories such as ligand RMSD, RMSF and Generalized Born Surface Area (GBSA) were generated to determine the binding free energy of the ligand in the two receptors (Figs 13 and 14 and Table 3).

Root mean square deviation The ligand RMSD and the protein backbone for each complex are shown over the course of simulation time, revealing a consistent deviation pattern exhibited by the two molecules throughout the simulation period (Fig 13). A low RMSD value indicates that the system is relatively stable, with relatively minimal deviation from the starting structure [59]. NPC204580 in complex with 3BAJ (Fig 13A) reached equilibrium at the start of the calculation at around ~5 ns with sidechain residuals fluctuating from time to time. Throughout the simulation time, oscillation was around ~2.0 Å, however between 45 ns and

**Table 2. Pharmacological features of the proposed compounds.**

| Pharmacokinetics | NPC204580 | NPC137813 |
|---|---|---|
| GI Absorption | High | High |
| BBB Permeant | No | No |
| P-gp substrate inhibitor | Yes | Yes |
| Cytochrome P450 (CYP) inhibitor | No | No |
| Log Kp (skin permeation) | -6.85 cm/s | -8.56 cm/s |
| Synthetic accessibility | 5.03 | 6.36 |

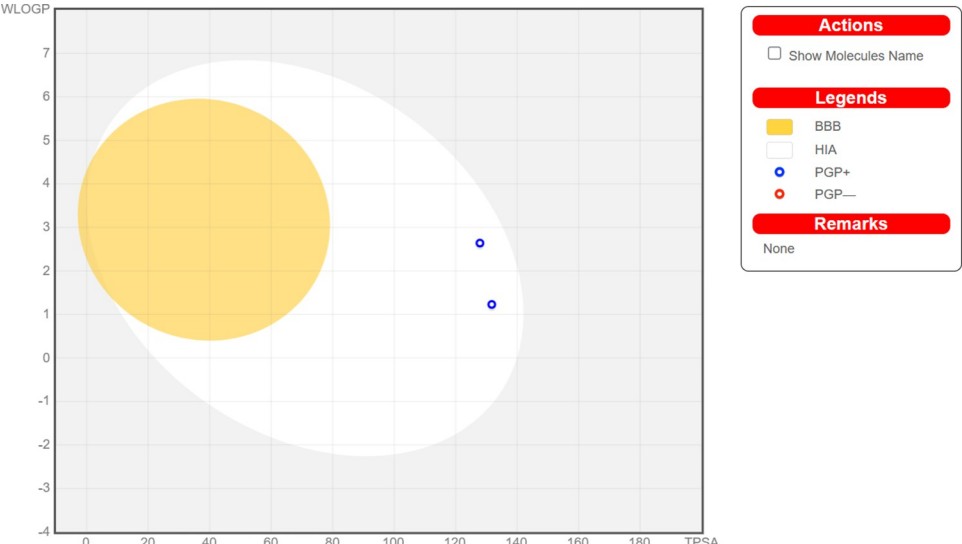

**Fig 11. BOILED-Egg plot for proposed molecules (NPC137813 and NPC204580).** The yolk-yellow region (Probable blood-brain barrier-BBB permeability), white region (Highly probable human intestinal absorption-HIA), grey zone (Low absorption and restricted brain penetration).

55 ns, there was a slight disparity as fluctuation rose up to 2.9 Å. This suggest a transient deviation from equilibrium and could be attributed to conformational changes or fluctuation in the complex. While NPC137813 in complex with 3BAJ receptor (Fig 13B) reached equilibrium at 15 ns, after which the complex exhibited a stable oscillation at an average RMSD value of ~2.5 Å throughout the simulation period. This suggests a possible stable equilibrium of NPC137813 in complexed with 3BAJ protein. NPC204580 in complex with 2QMJ (Fig 13C) was stable until approximately 73 ns into the simulation after which fluctuations were observed and stabilized at 87 ns. This is indicative of a transient destabilization followed by re-stabilization of the

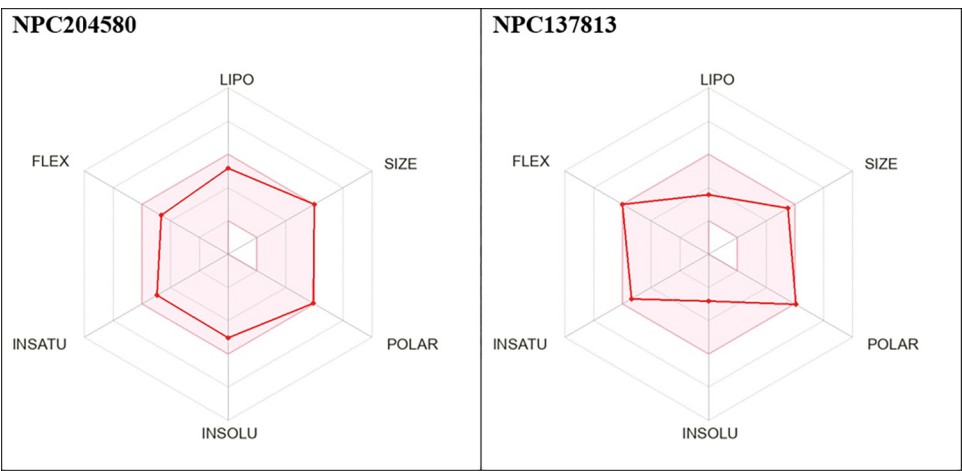

**Fig 12. The Bioavailability radar of the proposed ligands.** The pink region is a representation of the ideal range of properties of the two molecules, including Polarity (TPSA between 20 and 130 Å), Lipophilicity (log P between -0.7 and +5), Size (MW between 150 and 500 g/mol), Flexibility (no more than 9 rotatable bonds), Saturation (proportion of carbon atoms exhibiting sp3 hybridization no less than 0.25), and solubility, characterized by a logarithmic value of S (logS) not surpassing 6.

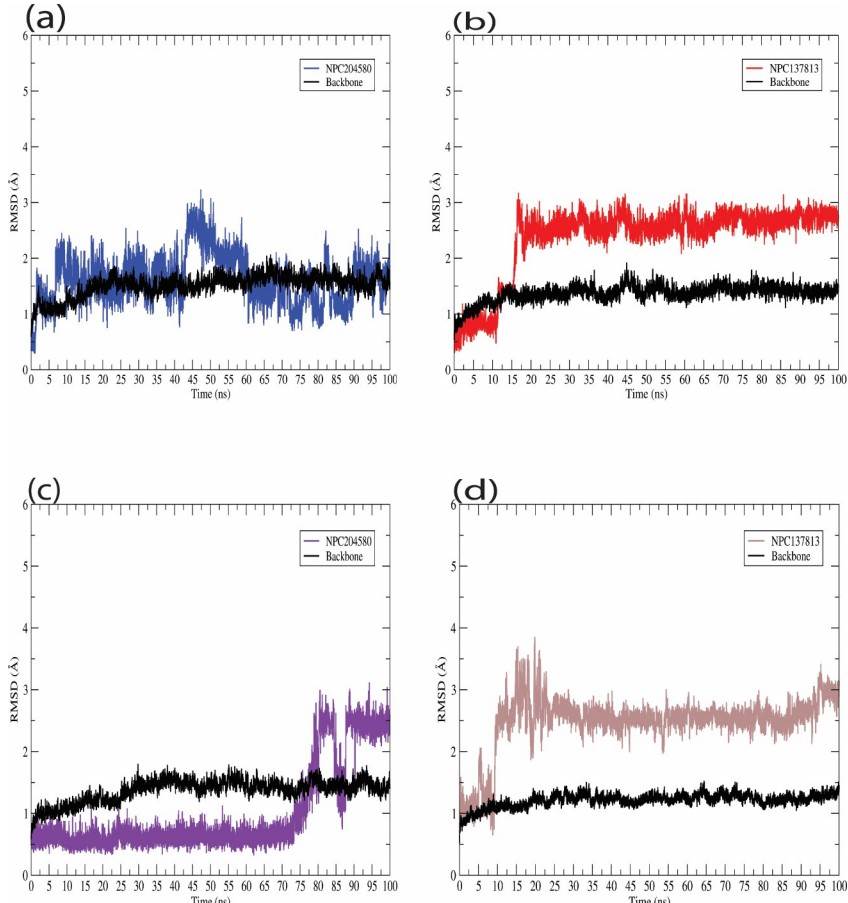

**Fig 13. Graphical representation of ligand RMSD of the proposed molecules and the protein backbone (black).** (a) NPC204580_3BAJ, (b) NPC137813_3BAJ, (c) NPC204580_2QMJ and (d) NPC137813_2QMJ.

complex. Conversely, NPC137813 in complex with 2QMJ (Fig 13D), exhibited initial fluctuation within the first 10 ns and attained equilibrium at ~11 ns that was maintained throughout the course of simulation time frame. This indicates that the ligand formed a stable complex with the receptor. Therefore, these findings enhance our docking results and dynamic behavior of NPC204580 and NPC137813 with the two receptors hence providing a valuable insight into their binding mechanism.

## Root mean square fluctuations

The stability of the protein-ligand complex in its dynamic state is heavily influenced by the fluctuations occurring in the backbone of a particular amino acid residue. The amino acid residues with low RMSF values are relatively more rigid than those with higher RMSF values [2]. To assess the stability of each amino acid in the two receptors, 3BAJ, and 2QMJ, when associated with the two proposed molecules, their RMSF value was computed (Fig 14). The protein's RMSF values, when complexed with the two molecules, exhibited comparable fluctuations during the whole duration of the simulation. The results revealed that in receptor 3BAJ (Fig 14A) only three amino residues exhibited a major fluctuation i.e. 240, 350, and 370 when bound with both NPC204580 and NPC137813, all the amino acid residues were below 10Å signifying that the protein-ligand complex was stable. Conversely, 2QMJ receptor in complex

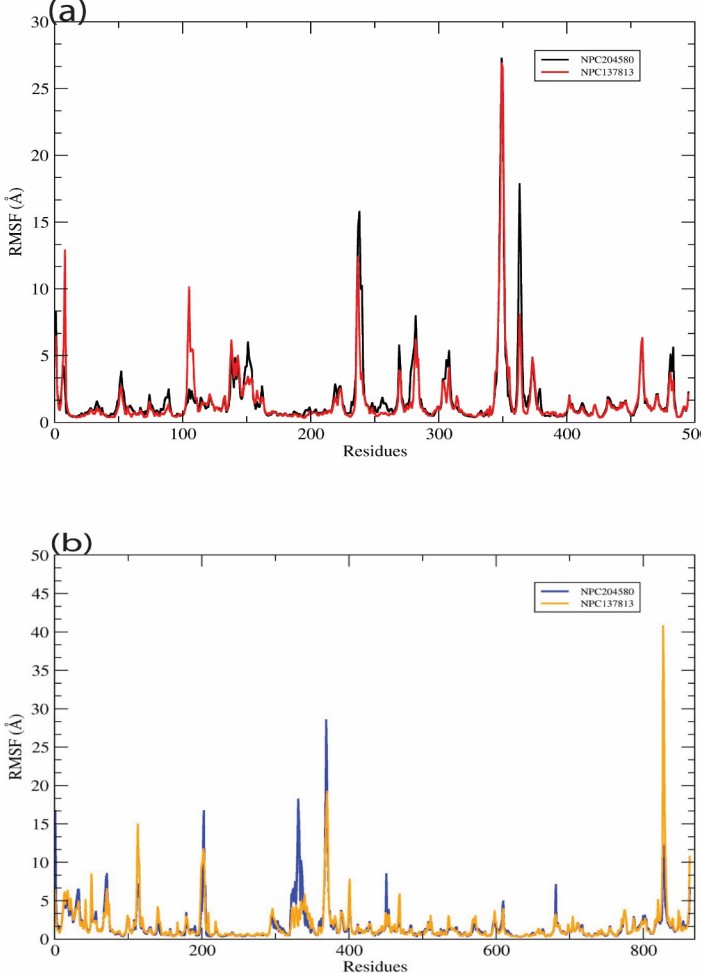

**Fig 14.** Graphical representation of the RMSF trajectories of individual amino acids of both the 3BAJ (a) and 2QMJ (b) receptors in associated with the proposed ligands (a) 3BAJ (NPC204580-black, NPC137813-red) and (b) 2QMJ (NPC204580-blue, NPC137813-yellow.

with NPC204580 and NPC137813 (Fig 14B) exhibited little fluctuation of its amino acid residues, except for four notable fluctuations at position 200, 340, 370, and 840. Among these residues, 840 exhibited a major fluctuation with a deviation of 42 Å, suggesting that the residue 840 experience a high degree of flexibility, possibly because of the residue being located on the flexible loop region of the binding site. However, the observed reduced fluctuations in the RMSF values provide clear evidence that the amino acids in both the 3BAJ and 2QMJ receptors effectively maintained their original conformational state throughout the duration of the molecular dynamics (MD) simulation. Furthermore, these amino acids effectively held the molecules, demonstrating a strong affinity and tight interaction.

## Free energy calculations

The GBSA is a commonly used approach for calculating protein-ligand free energies due to its excellent mix of computation precision and effectiveness [60]. In structure-based drug design, it is extensively used to determine end point binding free energy. MM/GBSA is much more precise and accurate as compared to most of the molecular docking [61]. In this study, for α-

**Table 3. Binding free energy (GBSA) of NPC137813 and NPC204580 in the two receptors.**

| Protein | Ligand | Binding affinity (ΔGbind) |
|---|---|---|
| α-Amylase | NPC204580 | -23.63 ± 0.21 |
| | NPC137813 | -32.87 ± 0.19 |
| α-Glucosidase | NPC204580 | -17.29 ± 0.15 |
| | NPC137813 | -23.39 ± 0.15 |

glucosidase and α-amylase, the GBSA offered a valuable insight into the strength of binding between these enzymes and the proposed molecules. Both molecules NPC204580 and NPC137813 exhibited a negative GBSA value in both enzymes (Table 3), indicating favorable binding affinity. In particular, NPC137813 portrayed greater binding interaction with both enzyme than NPC204580, i.e. ΔGbind -32.82 kcal/mol for α-amylase and ΔGbind -23.39 kcal/mol for α-glucosidase as compared to NPC204580 which showed a much lower binding interaction, i.e. ΔGbind -23.63 kcal/mol in α-amylase and ΔGbind -17.29 in α-glucosidase. NPC137813's higher binding affinity for both enzyme complexes suggests that it may be a more potent inhibitor than NPC204580. However, the two molecules exhibiting enhanced binding affinity to both enzymes could serve as promising therapeutic agents against P. hyperglycemia, pending in-vivo and in-vitro studies to validate computational finding.

## Conclusion

In conclusion, the objective of this study was to search for a potentially effective dual inhibitor for α-amylase and α-glucosidase from NPASS database utilizing AI techniques. A total of 30,926 molecules were subjected to multi-tiered virtual screening techniques. Ultimately, two drug candidate compounds were identified as possessing a notable propensity for binding to both 3BAJ and 2QMJ receptors relative to co-crystal standard ligand Acarbose. The two compounds exhibit pharmacokinetic profiles that are deemed acceptable, suggesting that they can be readily delivered to the intended site of action. The stability of these two molecules complexed with 3BAJ and 2QMJ were established to be consistence as demonstrated by MD simulation statistical metrics. Therefore, this study has employed some of these AI techniques to identify to molecules that can be prioritized for further evaluation as dual inhibitors of α-amylase and α-glucosidase. Therefore, the management of dietary sugar absorption and the regulation of post-meal rises in blood glucose levels can be achieved through the inhibition of these enzymes. However, additional experimental investigations (*in-vitro* and *in-vivo*) are required to substantiate the antidiabetic effectiveness of the proposed molecules.

## Supporting information

**S1 Fig. Pharmacokinetics properties of the screened molecules.**
(DOCX)

**S1 Table. Docking score and the interactions of selected compounds and acarbose with the α-amylase.**
(DOCX)

**S2 Table. Docking score and the interactions of selected top four compounds and acarbose with the α-glucosidase.**
(DOCX)

**S3 Table. Top ten docked poses of the selected molecules and the standard drug acarbose.**
(DOCX)

## Author Contributions

**Conceptualization:** Charles Otieno Ochieng, Jorddy N. Cruz, Cleydson B. R. Santos, Njogu M. Kimani.

**Data curation:** Wilberforce Ndarawit, Charles Otieno Ochieng, Njogu M. Kimani.

**Formal analysis:** Wilberforce Ndarawit, David Angwenyi, Cleydson B. R. Santos.

**Funding acquisition:** Njogu M. Kimani.

**Investigation:** Jorddy N. Cruz.

**Methodology:** Wilberforce Ndarawit, Charles Otieno Ochieng, David Angwenyi, Jorddy N. Cruz, Cleydson B. R. Santos, Njogu M. Kimani.

**Resources:** Jorddy N. Cruz, Cleydson B. R. Santos.

**Software:** Cleydson B. R. Santos.

**Supervision:** Jorddy N. Cruz, Njogu M. Kimani.

**Validation:** David Angwenyi.

**Writing – original draft:** Wilberforce Ndarawit.

**Writing – review & editing:** Charles Otieno Ochieng, David Angwenyi, Jorddy N. Cruz, Cleydson B. R. Santos, Njogu M. Kimani.

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
