## [Decision Letter · Decision Letter 0]

27 Sep 2024

PONE-D-24-24347Discovery of α-Amylase and α-Glucosidase Dual Inhibitors from NPASS Database for Management of Type 2 Diabetes: A Chemoinformatic ApproachPLOS ONE

Dear Dr. Kimani,

Thank you for submitting your manuscript to PLOS ONE. After careful consideration, we feel that it has merit but does not fully meet PLOS ONE’s publication criteria as it currently stands. Therefore, we invite you to submit a revised version of the manuscript that addresses the points raised during the review process.

**One or more of the reviewers has recommended that you cite specific previously published works. Members of the editorial team have determined that the works referenced are not directly related to the submitted manuscript. As such, please note that it is not necessary or expected to cite the works requested by the reviewer**

We look forward to receiving your revised manuscript.

Kind regards,

Mahmood Ahmed

Academic Editor

PLOS ONE

Journal Requirements:

Reviewers' comments:

Reviewer's Responses to Questions

**Comments to the Author**

1. Is the manuscript technically sound, and do the data support the conclusions?

Reviewer #1: Partly

Reviewer #2: Yes

Reviewer #3: Yes

Reviewer #4: Yes

2. Has the statistical analysis been performed appropriately and rigorously? 

Reviewer #1: No

Reviewer #2: Yes

Reviewer #3: N/A

Reviewer #4: Yes

3. Have the authors made all data underlying the findings in their manuscript fully available?

Reviewer #1: Yes

Reviewer #2: Yes

Reviewer #3: Yes

Reviewer #4: Yes

4. Is the manuscript presented in an intelligible fashion and written in standard English?

Reviewer #1: Yes

Reviewer #2: Yes

Reviewer #3: Yes

Reviewer #4: Yes

5. Review Comments to the Author

Reviewer #1: 1. Binding Score: The binding score may be presented as a line graph to show the scores of the to 10 or 20 poses from the docking. A single value of binding score (top pose) is not representative of the docking result.

2. ADMET and Drug-likeness screening: In light of the fact that the target enzymes are within the intestine, should high absorption be a consideration? Acarbose, the reference compound, is not well absorbed.

3. The MD should include the reference compound, acarbose, for good comparison.

Reviewer #2: Manuscript Title: Discovery of α-Amylase and α-Glucosidase Dual Inhibitors from NPASS Database for Management of Type 2 Diabetes: A Chemoinformatic Approach

Manuscript Number: PONE-D-24-24347

Authors: Wilberforce Ndarawit, Charles Otieno Ochieng, David Angwenyi, Jorddy N. Cruz, Cleydson B.R. Santos, Njogu M. Kimani, Ph.D

Journal: PLOS ONE

Summary

The manuscript titled "Discovery of α-Amylase and α-Glucosidase Dual Inhibitors from NPASS Database for Management of Type 2 Diabetes: A Chemoinformatic Approach" presents a study focused on the identification of potential dual inhibitors for α-amylase and α-glucosidase enzymes, which are relevant in the management of Type 2 Diabetes Mellitus (T2DM). The research utilizes computer-aided drug design (CADD) techniques to screen the Natural Products Activity and Species Source (NPASS) database, identifying two promising compounds: NPC204580 (Chrotacumine C) and NPC137813 (1-O-(2-Methoxy-4-Acetylphenyl)-6-O-(E-Cinnamoyl)-Beta-D-Glucopyranoside).

Evaluation

Originality and Significance:

The manuscript addresses a significant health issue, T2DM, by proposing a novel approach to identify dual enzyme inhibitors using chemoinformatic techniques.

The application of CADD to screen a large natural product database is innovative and demonstrates a modern approach to drug discovery.

Methodology:

The authors employ a rigorous and well-structured methodology, including virtual screening, molecular docking, and molecular dynamic simulations, to identify and validate potential inhibitors.

The study effectively combines various computational techniques to evaluate drug-likeness, pharmacokinetics, and toxicity profiles of the identified compounds.

Results:

The identification of NPC204580 and NPC137813 as potential dual inhibitors is well-supported by the computational data presented.

The binding affinities, stability assessments, and pharmacokinetic properties of these compounds suggest their potential efficacy in managing postprandial hyperglycemia.

Discussion and Conclusion:

The discussion is comprehensive, addressing the significance of the findings and comparing them with existing inhibitors.

The conclusion appropriately highlights the need for further in vitro and in vivo studies to validate the therapeutic potential of the identified compounds.

Presentation:

The manuscript is well-organized, with clear sections and logical flow.

Figures and tables are appropriately used to illustrate the computational results and support the conclusions.

Recommendation

Accept Without Corrections

The manuscript is well-written and presents a thorough and innovative study. The findings are significant and contribute valuable insights into the discovery of dual enzyme inhibitors for the management of T2DM. I recommend accepting the manuscript without any corrections.

Reviewer #3: The manuscript Discovery of α-Amylase and α-Glucosidase Dual Inhibitors from

NPASS Database for Management of Type 2 Diabetes: A

Chemoinformatic Approach. by Ndarawitet al is about the computer-aided drug design (CADD). Find below section wise comments

Title: Title of the work is attractive and informative, needs no change

Abstract: The work carried out by the authors are very much impressive, results obtained are quite interesting but unfortunately the work is not reflected from the abstract, I suggest authors to rewrite the abstract to make it more clear and self explanatory. Following un-necessary statements should be deleted from the abstract section

"Postprandial hyperglycemia, typical manifestation of Type 2 Diabetes Mellitus (T2DM)

characterized by elevated blood sugar levels, is associated with notable global morbidity and

mortality. Preventing the advancement of this condition by delaying the rate of glucose absorption

through inhibition of α-amylase and α-glucosidase enzymatic activities is of utmost importance.

Finding a safe antidiabetic drug is essential since the one that are currently on the market have

drawbacks like unpleasant side effects. Consequently, "

Language: The overall quality of the language is fine, however, there are some minor typo/syntax errors in introduction and in results Discussion section, the authors should carefully gone through the manuscript and correct the language issues

Introduction: This section is very well written, well managed and well organized, however the authors should use relevant and recent literature/citations to make the manuscript more attractive, after careful reading I suggest the following mandatory replacements

Replace reference number 3, 4, 5, 7 and 8 with

https://doi.org/10.3389/fendo.2023.1276836

doi: https://doi.org/10.1186/s12951-022-01715-z

doi: https://doi.org/10.1016/j.ejphar.2024.176356

doi: 10.1007/s00125-023-05992-7

doi: 10.1039/D3FO05456J

with the mentioned minor mandatory points,i recommend this manuscript for publication

Reviewer #4: The authors have conducted an interesting work which has good impact on field. The authors represented the work well and the results have been shown very clearly. I recommend it publication and to undergo the wet lab analysis to strengthen these findings.

6. PLOS authors have the option to publish the peer review history of their article (what does this mean?). If published, this will include your full peer review and any attached files.

Reviewer #1: **Yes: **Samuel Egieyeh

Reviewer #2: No

Reviewer #3: No

Reviewer #4: No

---

## [Author Response · Author response to Decision Letter 0]

7 Oct 2024

Response is attached on the uploaded file

---

## [Decision Letter · Decision Letter 1]

31 Oct 2024

Discovery of α-Amylase and α-Glucosidase Dual Inhibitors from NPASS Database for Management of Type 2 Diabetes: A Chemoinformatic Approach

PONE-D-24-24347R1

Dear Dr. Kimani,

We’re pleased to inform you that your manuscript has been judged scientifically suitable for publication and will be formally accepted for publication once it meets all outstanding technical requirements.

Kind regards,

Mahmood Ahmed

Academic Editor

PLOS ONE

Additional Editor Comments (optional):

Reviewers' comments:

Reviewer's Responses to Questions

**Comments to the Author**

1. If the authors have adequately addressed your comments raised in a previous round of review and you feel that this manuscript is now acceptable for publication, you may indicate that here to bypass the “Comments to the Author” section, enter your conflict of interest statement in the “Confidential to Editor” section, and submit your "Accept" recommendation.

Reviewer #2: All comments have been addressed

Reviewer #3: (No Response)

2. Is the manuscript technically sound, and do the data support the conclusions?

Reviewer #2: Yes

Reviewer #3: No

3. Has the statistical analysis been performed appropriately and rigorously? 

Reviewer #2: Yes

Reviewer #3: N/A

4. Have the authors made all data underlying the findings in their manuscript fully available?

Reviewer #2: Yes

Reviewer #3: Yes

5. Is the manuscript presented in an intelligible fashion and written in standard English?

Reviewer #2: Yes

Reviewer #3: No

6. Review Comments to the Author

Reviewer #2: The revised version and response letter have been examined critically. The authors have made the required changes and now the manuscript is fit for acceptance.

Reviewer #3: The authors have not incorporated the suggested points so I must reject this manuscript in its present form and encourage authors to to submit this manuscript in some other specialized journal

7. PLOS authors have the option to publish the peer review history of their article (what does this mean?). If published, this will include your full peer review and any attached files.

Reviewer #2: No

Reviewer #3: **Yes: **Wajid Rehman

---

## [Editor Report · Acceptance letter]

4 Nov 2024

PONE-D-24-24347R1 

PLOS ONE

Dear Dr. Kimani, 

I'm pleased to inform you that your manuscript has been deemed suitable for publication in PLOS ONE. Congratulations! Your manuscript is now being handed over to our production team.

Kind regards, 

on behalf of

Dr. Mahmood Ahmed 

Academic Editor

PLOS ONE